# Reliable and Accurate Wheel Size Measurement under Highly Reflective Conditions

**DOI:** 10.3390/s18124296

**Published:** 2018-12-06

**Authors:** Xiao Pan, Zhen Liu, Guangjun Zhang

**Affiliations:** Ministry of Education Key Laboratory of Precision Opto-Mechatronics Technology, Beihang University, Beijing 100083, China; panxiao0620541@163.com (X.P.); liuzhen008@buaa.edu.cn (Z.L.)

**Keywords:** High dynamic, reflective conditions, stripe image enhancement, wheel size measurement

## Abstract

Structured-light vision sensor, as an important tool for obtaining 3D data, is widely used in fields of online high-precision measurement. However, the captured stripe images can show high-dynamic characteristics of low signal-to-noise ratio and uneven brightness due to the complexity of the onsite environment. These conditions seriously affect measurement reliability and accuracy. In this study, a wheel size measurement framework based on a structured-light vision sensor, which has high precision and reliability and is suitable for highly reflective conditions, is proposed. Initially, the quality evaluation criterion of stripe images is established, and the entire stripe is distinguished into high- and low-quality segments. In addition, the multi-scale Retinex theory is adopted to enhance stripe brightness, which improves the reliability of subsequent stripe center extraction. Experiments verify that this approach can remarkably improve measurement reliability and accuracy and has important practical value.

## 1. Introduction

High-speed and high-precision vision measurement in outdoor environments has become an increasingly important tool for industrial scenes, e.g., vision perception [1], defect detection [2,3], and size measurement [4,5]. Among them, the online measuring instrument represented by a structured-light vision sensor is widely used, e.g., wheel size measurement [6,7], reconstruction of large forging parts [8], and monitoring of pantograph running status [9,10], where the online wheel size measurement has tremendous and remarkable practical meanings [11]. However, this measurement holds three main characteristics in the measuring process: (1) High speed, because the moving speed of the measured wheel is considerably fast, real-time calculation is required for the equipment; (2) high reflection, which is due to long time fraction, resulting in smooth wheel tread surfaces and specular reflection with active illumination, thereby causing overexposure (generally, the camera is set to working modes of a large aperture and short exposure time); and (3) irregular surface, that is, the topography of the measured wheel is irregular and the light is reflected away from the camera, generating low-brightness images. Regarding the restriction of sensor observation frustum, ensuring a good imaging status in the entire surface is difficult.

The imaging characteristics in highly reflective conditions are mainly reflected in dramatic brightness changes, that is, a high dynamic. The dynamic range (DR) of images represents the ratio of maximum and minimum brightness. A high DR indicates a large difference between the maximum brightness and minimum brightness of images. In dynamic images, the stripe brightness and width distribution are uneven. Figure 1 presents the camera imaging light path, where the measured wheel has high speed, high reflection, and an irregular surface; the camera is set to working modes of a large aperture, short exposure, and high sensitivity; and the stripe images captured have the characteristics of overexposure and uneven brightness. On the one hand, the stripe cannot be extracted in low-brightness segments. On the other hand, overexposed segments are not extracted correctly. The two phenomena in highly reflective conditions restrict the measurement reliability and accuracy of the structured-light vision system, especially in outdoor environments. In addition, current measurement systems of structured-light vision in outdoor illumination environments are vulnerable to external factors, such as an inaccurate trigger, vibration, surface material, light condition, and pollutants. These factors result in image unevenness, low signal-to-noise ratio (SNR), leakage, and false-positive cases, which seriously affect system reliability. In this case, we commonly decrease the exposure time for reducing the brightness of overexposed segments and increase exposure time for enhancing weak-lighting segments. Nevertheless, in practical applications, some contradictions exist between these aspects. For one aspect, the whole image is dimmed by decreasing the exposure time. Although the original bright segments are darkened, the dark segments become darker and may not even be detected. For another perspective, if the only exposure time is increased, thereby forming smears, then the original bright segments become brighter or even overexposed. Furthermore, some unused stripes may appear. The uneven brightness distribution, various widths, and low SNR of stripe images seriously affect the reliability and performance of the measurement system in highly reflective conditions. For example, in an online wheel size measurement system, when the strip image quality is low, high leakage and a false-positive rate can result, thereby seriously affecting the normal usage and reliability of the equipment. This case not only brings additional work for train inspectors, but also generates considerable security risks on normal train operation.

At present, experts have been investigating wheel size measurement sensors [12,13,14,15,16,17,18], and numerous types of products have been invented and applied, such as MERMEC [19], IEM [20], and KLD [21], which are suitable for low-speed trains. A stable and low-speed environment is selected to avoid measurement instability caused by outdoor high reflection. When the trains reach a high speed, the measurement accuracy seriously decreases. MERMEC adopts a specially designed high-resolution camera, and its measurement accuracy can reach 0.2 mm when the pass speed does not exceed 20 km/h. To date, existing measurement systems are mainly installed at the fence or garage entrance, under ideal environmental conditions. However, systems [17] installed in the main line are seriously affected by outdoor lighting and other factors. Not one measurement system of structured-light vision is robust and reliable and meets the requirement of online measurement in highly reflective conditions. Therefore, the quality of stripe images in highly reflective conditions must be improved to increase system reliability. To our knowledge, no solution has been proposed to improve the measurement system of structured-light vision in highly reflective conditions. The conventional methods often focus on image brightness enhancement. Recently, various image enhancement methods have been proposed to improve low-brightness images. These methods consist of spatial and frequency domains. Frequency methods mainly address the transform coefficients in a certain frequency domain of the image, and then the enhanced image is obtained by inverse transformation. In Reference [22], a multi-resolution overlapping sub-block equalization method based on wavelet is proposed. The image is decomposed and equalized and then reconstructed through inverse wavelet transform, but the method is unsuitable for images in the histogram concentrated distribution. The gray field transformation method adjusts the DR or contrast of the image, which is an important means of enhancement. This method transforms the grayscale in the original image into a new value, which can increase the gray DR. The categories of transformation function forms include linear, piecewise linear, and nonlinear. For the nonlinear gamma correction method [23,24], the selection of a nonlinear correction function is particularly important, its local processing capacity is poor, and the effect on structured-light image enhancement is not evident. In addition, histogram equalization [25] can increase the uniformity of gray distribution and enhance image contrast. However, this method does not consider the frequency and details of the image, which is likely to cause overenhancement. The gradient enhancement method [26] converts the image to the logarithm domain with gradient processing, thereby reducing the gradient value for compressing the image DR and increasing the large gradient to enhance edges in the image. Nevertheless, this method is unsuitable for real-time application. The Retinex [27] enhancement method utilizes the Gaussian smoothing function to estimate the luminance component of the original image. This method is suitable for handling images with low local gray value, thereby effectively enhancing the details of the dark part and maintaining the original image brightness to a certain extent while compressing the image contrast. In recent years, numerous forms based on Retinex theory, e.g., single scale [28], multi-scale [29], and color image enhancement [30], have achieved the ideal enhancement effect.

The accuracy of stripe center extraction determines the measurement precision, whereas the stripe quality affects the accuracy of stripe center extraction. Therefore, this paper presents a method for wheel size measurement in highly reflective conditions. First, the stripe imaging process in highly reflective conditions is analyzed. Second, the stripe quality evaluation is proposed to locate enhanced stripe segments accurately. The image brightness is enhanced based on multi-scale Retinex (MSR). Physical experiments proved that, in the outdoor complex environment of the railway, the proposed method, compared with the existing methods, can improve the accuracy of the online wheel size measurement system and effectively reduce leakage and false-positive rates.

The remainder of this paper is organized as follows. Section 2 briefly reviews the wheel profile reconstruction and overviews the scheme of high-dynamic image processing. Section 3 describes the stripe imaging in highly reflective conditions and presents an effective enhancement approach for a high-dynamic stripe image. Section 4 discusses the experiments and evaluations. Section 5 draws the conclusion.

## 2. System Overview

The online wheel size dynamic measurement system is composed of equipment on rail, control unit, data processing, and wheel size calculation modules. Equipment on rail includes two structured-light vision sensors, which are installed under a measured wheel. The control unit receives the wheel arrival signal from a magnetic trigger and generates data synchronously to cameras and lasers. Data processing involves image acquisition, image processing, stripe center extraction, and 3D reconstruction, which facilitate wheel size calculation. Wheel cross profile acquisition is a critical process in measurement, and its layout is shown in Figure 2. Two structured-light vision sensors are installed inside and outside the measured wheel, and their laser planes are adjusted to be coplanar in space and pass through the wheel center.

When the wheel passes the trigger position, the inner and outer sensors observe the portion of the wheel cross profile. Among them, the inner camera is mainly used to detect the inner rim and part flange, and the outer camera is utilized to detect the tread and part flange. Let $K_{i},K_{o}$ be the inner and outer camera intrinsic parameters, $F_{i},F_{o}$ be plane functions in a camera coordinate system (CCF), and $R_{oi},T_{oi}$ be transformation matrices of the outer camera to the inner camera. The sensors are calibrated offline [31,32]. Let $C_{i}$ and $C_{o}$ be the wheel profiles from the inner and outer sensors. After translating $C_{o}$ to the inner CCF, $C_{o}^{\prime }=R_{oi}\cdot C_{o}+T_{oi}$, the complete wheel profile is $C_{c}=C_{i}\cup C_{o}^{\prime }$, which is used to calculate the parameters in accordance with the wheel size definition.

Specifically, we focus on solving the stripe image enhancement in a wheel size measurement system, the diagram of which is shown in Figure 3. Figure 3 illustrates that stripe skeleton extraction is implemented on the raw images captured by each camera. Based on the stripe quality evaluation criteria, the stripe segments to be enhanced are established along the stripe skeleton trajectory. Based on MSR theory, the stripe with low-brightness segments are enhanced for center extraction.

The reliable and accurate wheel size measurement framework can be concluded as follows: **Step** **1.**Stripe images are synchronously captured from sensors installed around the measured wheel.**Step** **2.**Image dilation and low threshold segmentation are adopted to obtain the stripe area in which valid stripes are located and the central skeleton points are extracted.**Step** **3.**Stripe image quality evaluation criteria are established in accordance with the center extraction principle.**Step** **4.**Based on the stripe quality criteria, the stripe segments with low brightness are obtained along the stripe skeleton trajectory.**Step** **5.**MSR brightness enhancement is implemented to the low-quality stripe segments generated in step 4. Moreover, the valid enhanced stripe is segmented accurately based on the reflectivity.**Step** **6.**The center points of the enhanced stripe are extracted and the wheel cross profile is reconstructed. Finally, wheel size parameters are calculated.

## 3. Algorithm Implementation

In this section, the characteristics of wheel imaging and the cause of low brightness are analyzed. High-dynamic stripe enhancement includes stripe quality evaluation, locating enhanced segments, stripe enhancement, and enhanced stripe segmentation.

### 3.1. Stripe Imaging Model and Analysis

#### 3.1.1. Stripe Imaging Description

The structured-light vision sensor projects a laser onto the measured object surface and obtains the image of the reflected laser line. The camera-sensitized illumination model can be expressed as $I=I_{e}+I_{d}+I_{s}$, where $I_{e}$ is the ambient component, $I_{d}$ denotes the diffuse component, and $I_{s}$ represents the specular component. $I_{e}$ is independent of the viewing angle and related to the light source and object reflectivity, which is negligible in structured-light measurement due to an optical filter. $I_{d}$ is determined by the view angle and object surface reflectivity. When the view angle is small, the diffuse reflection component received is considerable. Meanwhile, the view angle, object reflectivity, and specular direction determine $I_{s}$. When the angle between the view vector and specular direction decreases, the specular component increases and the stripe is prone to overexposure. Therefore, in the structured-light imaging process, we attempt to avoid the effect of the specular component while increasing the diffuse component to improve the image SNR and uniform brightness and ensure measurement accuracy.

Figure 4a,b shows the stripe images captured by the outer and inner cameras, respectively. After the line laser is irradiated to the wheel surface, most of the parts, except those absorbed by the wheel, are reflected and composed by $I_{d}$ and $I_{s}$. Most of the laser energy is reflected away due to the high reflectivity of the wheel surface, large changes in topography, and the imaging angle of the camera, resulting in partial darkness of the stripe image, as shown in Figure 4a r1 and r2 segments. Similarly, for the large view angle of the inner camera and the reflectivity of the wheel inner side, the stripe brightness of Figure 4b r1 and r2 segments are dark in the inner rim.

Figure 5 displays the gray value distribution in the stripe and normal directions. Ideally, the gray distribution is even in the stripe direction and is inclined to a Gaussian distribution in the normal direction. However, the grayscale distribution of the image captured on the site changes strongly. Figure 5b shows that the gray distribution in the stripe direction and the curve fluctuate severely and are accompanied by two dark curve segments marked with a red dotted frame. Figure 5c illustrates the gray distribution curves in a normal direction at approximately eight sampling points, presenting as spiked, steep, and narrow. If we increase the camera exposure time, then the bright segments become brighter, which leads to overexposure. Therefore, the best method is to enhance the brightness of dark segments and maintain the original ones.

Figure 6 displays the online stripe images captured from the outer cameras. Three different segments are used, namely, low brightness, overexposure, and stray light. Overexposure and other stray lights that are unavoidable in real measurement are reduced by decreasing the exposure time. Generally, the low-brightness stripe image is the main factor causing the leakage, which is the key focus of this study.

Figure 7 shows the center extraction results of stripe images. Figure 7a displays the images with low brightness in tread segments. Figure 7b illustrates the stripe images with overexposure in the flange segments. The former cases cause useful segments to not be extracted accurately. By contrast, the key segments in the tread are almost not extracted, which causes wheel parameters to not be calculated.

#### 3.1.2. Stripe Brightness Analysis

When the incident light is a laser, the wheel surface is irregularly undulating even though the light is regular. Therefore, the reflected light shows irregularity with bright, dark, or uneven distribution. Let $\alpha_{1}$ and $\alpha_{2}$ be the incident and scatter angles, respectively. Then, the scatter light intensity is described as:(1)$$\{\begin{array}{l} I_{s}(x,y)=I_{0}(x,y)\exp(-\sigma_{\varphi }^{2})\\ \sigma_{\varphi }=\frac{2\pi }{\lambda }\sigma_{h}(\cos\alpha_{1}+\cos\alpha_{2}) \end{array},$$
where $I_{0}(x,y)$ represents the incident light, $\sigma_{h}$ denotes the surface roughness, and λ refers to the wavelength.

In the online wheel size dynamic measurement system, the near-infrared laser wavelength is λ=808 nm, and the wheel surface roughness is $\sigma_{h}\in [40,100]$. Figure 8 shows that the abscissa is the ratio of the surface roughness to the wavelength, and the vertical axis is the normalized intensity of scattered light. The scattered light increases as the ratio of the surface roughness to the wavelength increases and decreases as the view angle increases. Given that the relative roughness of the surface is small, it can be regarded as a fixed value. Figure 8 S1 illustrates that the surface is smooth, and the scattered light intensity is weak. The angle of observation and surface normal vector are large, and the scattered light is weak. Figure 8 S2 indicates that the stripe observed by the camera becomes dark after a long period of friction of the tread area, whereas other rough areas become bright.

Figure 4a exhibits that the stripe in the tread area is darker than other rough areas due to smoothness. By contrast, Figure 4b shows a dark area in the inner rim. In the stripe image captured by the outer camera, as shown in Figure 4a, tread segments of r1 and r2, due to the smooth surface and large view angle, allow minimal light into the digital sensor. The reflectivity of the stripe image from the inner camera, due to oil pollution above the inner rim, is reduced and the stripe is dimmed in Figure 4b r1 and r2 segments.

### 3.2. High-Dynamic Stripe Image Processing

#### 3.2.1. Quality Evaluation of Stripe Image

A good laser stripe is a prerequisite for structured-light vision sensor measurement. Therefore, laser stripe image quality evaluation is conductive for determining whether or which stripe segments need to be enhanced. Xie Fei [33] conducted a statistical behavior analysis for the laser stripe center detector based on Steger algorithm [34]. The quantitative relationship between the center point uncertainty and its surrounding SNR is determined. Based on the stripe extraction algorithm, the stripe line satisfies the Gaussian distribution in a normal direction, and its center coordinate is the laser energy point. The features of the ideal stripe include a uniform brightness in the stripe direction, a Gaussian distribution in the normal direction, and space continuous. Consequently, three aspects are considered, namely, gray distribution in the stripe direction, gray distribution in the normal direction, and the space continuous.

a. Stripe direction

Let $g_{i}$ be the brightness of each stripe pixel, $\mu_{g}$ be the mean value, and $\sigma_{g}$ be the variance. Generally, a high $\mu_{g}$ and small $\sigma_{g}$ indicate that the stripe has uniform brightness without fluctuations. In other words, the overall quality is good. Based on experience, the image with $\mu_{g}>0.8$, $\sigma_{g}<0.2$ is a good stripe image.

b. Normal direction

Based on the center extraction algorithm, the gray in the normal stripe direction meets the Gaussian distribution, which induces a small positioning error. We determine the difference between the origin and Gaussian-filtered stripes to describe the normal gray distribution. The small difference proves that the stripe conforms to the Gaussian distribution. We set the difference (namely, image noise), $\rho_{i}=\sum \left\|h_{i}-h_{i}⊗G_{\mu -\sigma }\right\|$, and calculate the mean, $\mu_{\rho }$, and variance, $\sigma_{\rho }$, where $h_{i}$ is the normal sample gray and $G_{\mu -\sigma }$ represents the Gaussian convolution mask. A small $\mu_{\rho }$ and $\sigma_{\rho }$ manifest few burr and uneven parts, which indicates a good quality. The template size of the selected Gaussian filter should refer to the actual width of the stripe. Generally, we set $\mu_{\rho }<0.024$, $\sigma_{\rho }<0.008$ as the evaluation criteria in the normal direction.

c. Stripe continuity

A good laser stripe should be continuous in space. Correspondingly, when the disconnection segment is detected, image enhancement is implemented. We set $D_{p}$ as the percentage of the effective stripe in the entire stripe. Let the distance between the adjacent stripe points be $d_{i}$, the mean distance of $d_{i}$ in the entire stripe be $\mu_{d}$, and the variance be $\sigma_{d}$. Similarly, a small $\mu_{d}$ and $\sigma_{d}$ indicates that the stripe is complete and with few disconnected parts. Usually, when adjacent stripe points’ $\mu_{d}<5pixel$, it is valid for measurement. Furthermore, this parameter is determined by the real measurement requirement.

In addition, $\mu_{i}$ and $\sigma_{i}$, i=g,ρ,d are normalized in [0,1]. Regarding the stripe quality, the highest priority is stripe continuity, which ensures measurement reliability. Then, the gray value in the stripe direction and the noise in the normal direction determine the measurement accuracy.

#### 3.2.2. Enhanced Stripe Segments Location

Based on the stripe quality evaluation criteria described in Section 3.2.1, we conduct stripe quality evaluation to enhance the stripe segments. Therefore, to obtain the coordinates of the stripe image center roughly, the stripe is segmented out, and binary operation and skeleton extraction are implemented, thereby obtaining the continuous stripe points along the stripe direction. Figure 9 shows the establishment process of the stripe quality regions from ① to ④. Quality evaluation is implemented in each stripe point in accordance with evaluation criteria; thus, we can obtain low-quality segments for brightness enhancement and extract good-quality segments for avoiding additional calculations.

Figure 9 displays the images captured by the outer and inner cameras, where ① is the original image whose brightness is uneven and contains good- and low-quality segments. ② presents the segmented and binarized image from the original image. ③ shows the stripe skeleton points extracted from ②. In ④, the red points indicate the stripe segments with good quality, and the green dot rectangle region is the segment to be enhanced, e.g., segments R1 and R2 in the tread and flange regions and R3 in the rim.

#### 3.2.3. Image Brightness Enhancement

Retinex theory was introduced by Edwin. H. Land [32], who found that the color perceived by the human vision system is determined by the reflection of an object and does not involve the light from a scene. Based on this analysis, the stripe image brightness depends mainly on the surface reflectivity. Hence, stripe brightness enhancement based on Retinex is appropriate for solving high-reflection problems. The Retinex algorithm that holds the image is composed of two parts, namely, the reflected and incident components. To acquire the reflection component further to restore the original appearance of the object, the illuminance component is obtained by computing the brightness between pixels. The image can be expressed as follows:(2)I(x,y)=R(x,y)⋅L(x,y),
where L(x,y) represents the incident light and R(x,y) denotes the reflection property of the object. I(x,y) indicates the image to be enhanced. In fact, the incident light L(x,y) directly determines the DR of a pixel reaching in the image. R(x,y) determines the intrinsic nature of the image. The purpose of Retinex theory is to obtain R(x,y) from I(x,y). Equation (2) is transformed into the logarithmic domain: (3)log(I(x,y))=log(R(x,y))+log(L(x,y)).

Land proposed a center/surround Retinex algorithm with the basic idea that the brightness of each center pixel is estimated by setting different weights around the pixel. Jobson finally determined that the Gaussian surround function can achieve good results. Equation (3) can be expressed as:(4)log(R(x,y))=log(I(x,y))−log(F(x,y)⋅I(x,y)),
where the Gaussian function,$F(x,y)=\frac{1}{2\pi \sigma^{2}}\exp(-\frac{\left(x^{2}+y^{2}\right)}{2\sigma^{2}})$, and σ refers to the scale parameter, which directly affects the estimation of the incident component. When σ is small, the Gaussian template is small and the Gaussian function is relatively steep. The estimated component of the incident after convolution is also relatively rugged, with a strong dynamic compression. However, the detail is retained and the brightness is poor. On the contrary, when σ is large, the Gaussian template is large and the Gaussian function is relatively gentle; the convolution of the incident component is also relatively smooth and the performance of brightness fidelity is good. However, the dynamic compression ability is poor and the details of the enhancement are not evident. Thus, the Retinex algorithm cannot guarantee detail and brightness enhancements based on a single σ. The MSR is a generalization of the single-scale Retinex algorithm, which ensures detail and brightness enhancement:(5)$$r(x,y)=\sum_{j=1}^{K}W_{j}\{\log I(x,y)-\log[I(x,y)\cdot F_{j}(x,y)]\},$$
where r(x,y)=log(R(x,y)), K is the total number of σ, $W_{j}$ signifies the weight, and $\sum_{j=1}^{K}W_{j}=1$. Under normal circumstances, MSR adopts high, medium, and low scales, that is, K=3, $W_{1}=W_{2}=W_{3}=1/3$. When the reflection is computed, the enhanced pixels are mapped by normalizing its corresponding reflection to [0,255]. The pseudocode of high-dynamic stripe image enhancement is shown in Algorithm 1 as follows:
**Algorithm 1.** High Dynamic Stripe Image Enhancement**Input:** Low brightness stripe image ***I***.**Output:** High quality stripe image ***I**_e_*. 1: **Function stripeImgHandler** 2: Compute surface reflectivity ***f***; 3: Normalize ***f*** to gray value and get enhanced image ***I**_t_*; 4: Calculate the initial vaue of segmentation threshold *T*; 5: *T* = 0.5 × (min(*T*) + max(*T*)) 6: Set segment flag = false; 7: **While**(!flag) **do** 8: Find the image index, *g* = *find*(***f*** > *T*); 9: Calculate stripe area segmentation threshold, 10: *T_n_* = 0.5 × (*mean*(***f***(*g*)) + *mean*(***f***(~ *g*))) 11: flag = *abs*(*T* − *T_n_*) < 0.1; 12: *T* = *T_n_* 13: **End while** 14: Segment ***I**_t_* with *T*; 15: return ***I**_e_*; 16: **end function.**

Figure 10 shows the stripe images before and after enhancement. Figure 10a presents the original image with uneven luminance distribution. In particular, the brightness of the middle region (which is used to calculate the tread wear) is dark, which easily affects tread-base point extraction. In Figure 10b, the tread area is recovered, thereby ensuring the accuracy of stripe center extraction.

#### 3.2.4. Stripe Segmentation

The enhanced image often contains an enhanced background and stray stripes; thus, segmenting the effective stripe is necessary. This study does not directly use the image segmentation method based on the gray threshold because the image brightness changes greatly and is difficult to segment properly. Nevertheless, the reflection of the stripe area is distinct from the background. We adopt reflectivity to generate a valid stripe mask for filtering out the enhanced stripe. Figure 11a,b shows the gray and reflectivity distributions, respectively. The gray image is not considerably different in the background. By contrast, the reflectivity distribution is evident, and the stripe section can be segmented accurately.

## 4. Experiments and Evaluations

### 4.1. Experimental Setup

To verify the effectiveness of this proposed method, we applied it to the online wheel size measurement system installed in Lixian County, Hebei Province, China. Figure 12 shows that four structured-light vision sensors are equipped under the rail, and each wheel is measured by two structural light sensors mounted inside and outside the rail. Given that this device is installed outdoors and in the main line, complex lighting with various interferences, complicated wheel appearance, and reflectivity cause the collected images to show high-dynamic characteristics, especially in the wheel tread. Table 1 lists the sensor configuration parameters.

### 4.2. Comparison of Stripe Image Processing Results

The processing methods include the proposed method (PM), histogram equalization (HE), contrast adjustment (CA), logarithmic transformation (LT), convolution filter (CF), median filtering (MF), and sharpening operation (SO). The abbreviations for each method are utilized to facilitate the description.

#### 4.2.1. Comparison of Stripe Quality

To verify the improvement of the stripe image, its quality before and after the enhancement treatment was measured. We select four data sets; each contains 100 images, where data sets 1, 2, 3, and 4 present 40%, 50%, 60%, and 70% low brightness stripes (induced by a highly reflective object surface), respectively. The image was enhanced by applying the abovementioned methods and the stripe quality evaluation parameters are calculated. Table 2 shows the statistics of image quality evaluation parameters for different data sets corresponding to different enhancement methods. The optical stripe quality evaluation parameters corresponding to the proposed method are the best in different datasets, indicating that the proposed method can effectively improve the stripe quality under highly reflective conditions.

#### 4.2.2. Time Statistics

To verify the processing efficiency of each method, the calculation time of different processing steps is separately calculated. The computer configuration is a 32-bit Windows 7 operating system, with a 3.1 GHz CPU and 8 GB memory. Table 3 lists the time consumption of each process. To simplify the expression, SD, SE, SS, SCE, TBA, and AC stand for stripe detection, stripe enhancement, stripe segmentation, stripe center extraction, total before acceleration, and after acceleration, respectively.

We adopt multi-thread flow acceleration and a serialized output program, which can theoretically reach unlimited speed. However, when considering the hardware overhead, including thread switching and additional calculation, the actual processing speed based on 13-level acceleration can increase to 30 HZ with the train speed of up to 189 km/h. Although the proposed method induces numerous additional calculations, it can still meet the measurement needs of high-speed trains through multi-thread flow acceleration, which meets the online testing requirements of normal freight and passenger trains. Furthermore, the acceleration framework is versatile and suitable for other image processing methods.

#### 4.2.3. Comparison of Various Image Enhancement Methods

To verify the effectiveness of the proposed method, we compare various enhancement methods. Figure 13 and Figure 14 show the inner and outer stripe image enhancement results, respectively. Each row contains four types of images to be processed and presents the processing result of an image enhancement method. Figure 13 and Figure 14 illustrate that this method has a greater improvement than other methods. By contrast, the HE method leads to serious stripe overexposure. The CA method adjusts all pixels’ gray values, but the brightness of the enhanced part remains relatively low. LT, MF, and SO do not play a great role in brightness enhancement.

Figure 15 and Figure 16 illustrate the outer and inner stripe image center extraction results based on different methods. However, the center extraction results of other methods are unsatisfactory; thus, it cannot be applied to actual measurement. Figure 15c,f presents numerous stay stripes extracted. In Figure 15d,e,g,h, the stripe is incomplete for measurement. In Figure 15b, the entire stripe is extracted correctly, especially the tread area. Correspondingly, the same result exists in Figure 16, that is, only the proposed method can guarantee the integrity and accuracy of stripe extraction.

Given that the coordinates of the center point obtained by other methods differ greatly from the stripe required for actual measurement, contrasting is not needed. In this study, only the reconstructed result corresponding to the original image is compared with the proposed method. Figure 17 shows the wheel cross contour reconstruction results without image processing. Some parts are missing in stripe images, including the tread and inner rim, causing the last wheel contour to be incomplete and to not be utilized for calculation. On the contrary, Figure 18 illustrates that all stripe images are enhanced properly and the last wheel cross contour is complete; thus, the original missed wheels can still be measured accurately.

### 4.3. Performance Evaluation of Online Measurement

For evaluating the online measuring performance of the proposed method, we repeated the experiment with the standard wheel, performed comparative experiments under highly reflective conditions, and conducted onsite dynamic measurement. As we cannot determine the precise size of each wheel, the statistical leakage and false-positive rates are mainly explained.

#### 4.3.1. Measurement Accuracy

The standard freight wheel is adopted as a measured object whose size is identified. We implement 100 repeated-experiments under various highly reflective conditions (highly reflective wheel moving back and forth in the measuring process) and perform statistics of three key wheel size parameters, including flange thickness, tread wear, and rim thickness. Table 4 presents the measurement accuracy of the wheel size by using different methods. Only the results of PM, HE, CF, and SO methods are calculated. Furthermore, the measurement accuracy of the proposed method is considerably higher than that of other methods.

#### 4.3.2. Detection Rate Statistics

To evaluate the robustness and accuracy of the proposed method, an equipment installed in Shuohuang railway, Hebei, China, is selected as the experimental subject. We analyze the leakage and false-positive rates from 00:00:00 to 13:00:00 on 12 November 2017. A total of 56 vehicles and 23,436 wheel sets pass through the measurement equipment.

In practical application, the inspection department is most concerned about the reliability and accuracy of the equipment during operation. The leakage rate is used to evaluate the reliability of the equipment and is a prerequisite for an accurate measurement. The false-positive rate is utilized to check the accuracy of the equipment, which can effectively improve the inspector efficiency. Figure 19 illustrates the leakage and false-positive rates with different methods in an outdoor complex environment, respectively.

Figure 19a shows that only SO, CF, and the proposed method can effectively reduce the leakage rate. As other methods have limited brightness enhancement of the original image, the leakage rate is not considerably improved. In comparison with SO and CF, the proposed method can reduce the leakage rate to 0.38%. Moreover, in comparison with the false-positive rate shown in Figure 19b, the SO and CF methods are considerably greater than 0.14%, which is achieved by the proposed method. Therefore, the proposed method can effectively reduce the leakage and false-positive rates and can meet the requirements of the railway inspection department.

## 5. Conclusions

High speed, dynamic, and precise laser vision sensors are widely used in online measurement in a complex environment, for example, an online wheel size measurement system. However, the stripe images show high-dynamic characteristics of low SNR due to onsite highly reflective conditions, which seriously affects the measurement accuracy and reliability. To address this issue, a high-precision and high-reliability wheel size measurement system suitable for highly reflective conditions was proposed in this study. Initially, the stripe image quality evaluation mechanism was established and used to assess which part of the stripe was enhanced. MSR theory was adopted to enhance the stripe brightness, and reliable stripe segmentation was achieved through reflectivity, thereby solving the problem that low-brightness stripes are not extracted in reflective conditions. Physical experiments analyzed the before and after enhancement results, thereby proving that the method can effectively improve the measurement reliability and greatly reduce the leakage and false-positive rates. The method has great practical value for improving the accuracy and reliability of a wheel size measurement system in highly reflective conditions. However, some space remains for promotion; as we review the enhanced images, the width of the stripe changes slightly. Therefore, research on a multi-scale stripe center extraction algorithm is necessary, in which case the stripe center points can be extracted adequately. Furthermore, for a high measurement frequency, we plan to utilize an FPGA or GPU hardware equipment to improve the stripe processing and center extraction speed, which renders it suitable for a wide range of applications. For some cases, such as the stray stripe images with extra high reflectivity and overexposure shown in Figure 7a, improvement was impossible. The proposed method was the best approach for decreasing exposure and rejecting stray stripes. Similarly, no laser was captured in tread segments in Figure 7b, and our method also cannot restore stripes, which is unavoidable.

## Figures and Tables

**Figure 1 sensors-18-04296-f001:**
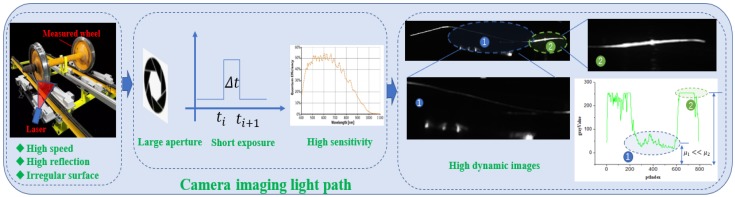
Camera imaging light path of the online wheel size measurement system.

**Figure 2 sensors-18-04296-f002:**
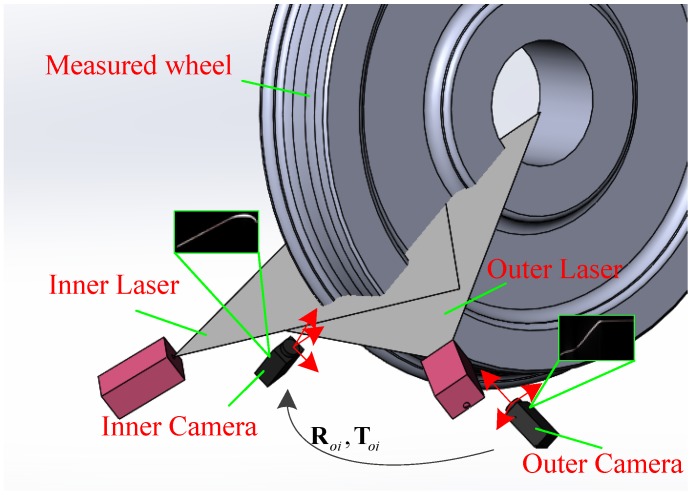
Layout of wheel profile acquisition sensors.

**Figure 3 sensors-18-04296-f003:**
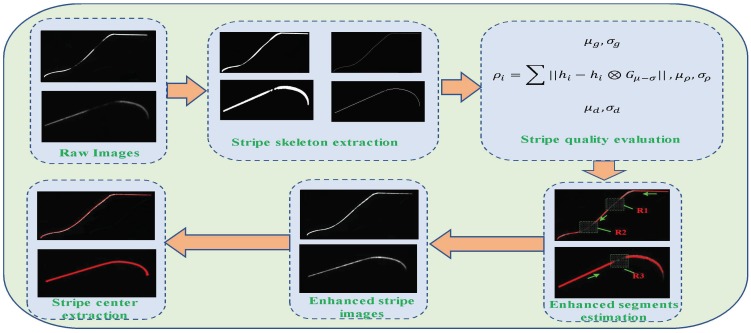
Diagram of stripe image enhancement in highly reflective conditions.

**Figure 4 sensors-18-04296-f004:**
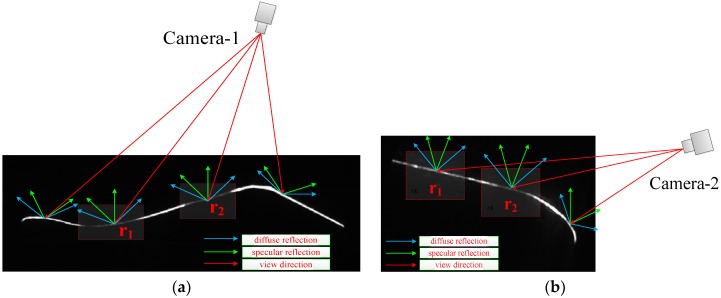
Laser brightness perception of stripe images captured by outer and inner cameras. (**a**) Laser brightness perception of the image from the outer camera; (**b**) laser brightness perception of the image from the inner camera.

**Figure 5 sensors-18-04296-f005:**
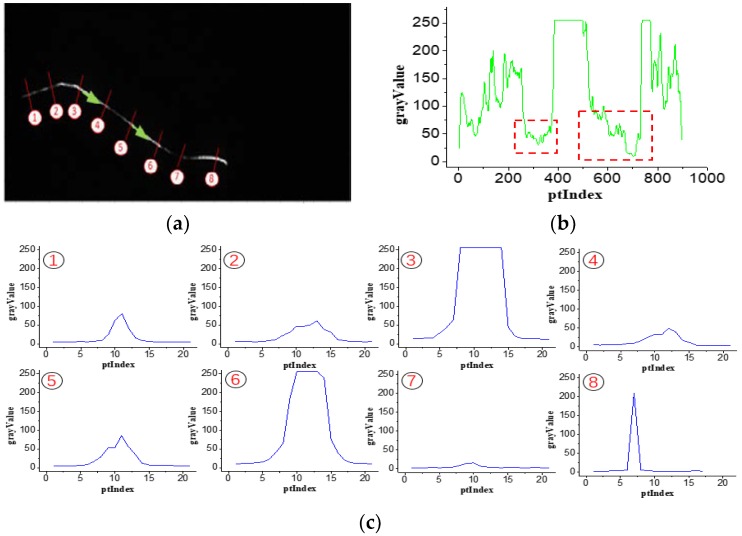
Gray distribution of the stripe image in the stripe and normal directions. (**a**) Stripe image from the outer camera. (**b**) Gray value distribution in the stripe direction. (**c**) Gray distribution curves of the stripe image in the normal direction.

**Figure 6 sensors-18-04296-f006:**
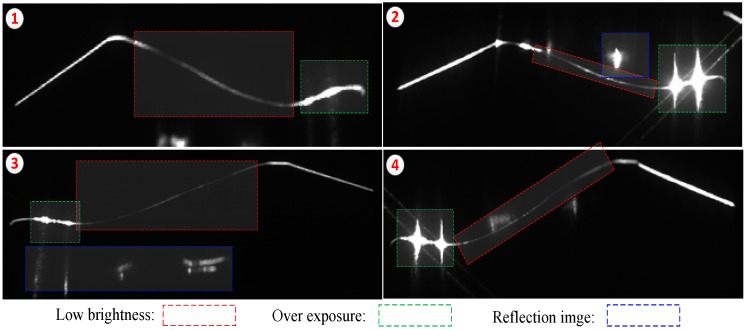
Online stripe images captured from the outer camera. ① and ③ present the stipe images captured with low exposure. ② and ④ exhibit the stipe images captured with high exposure.

**Figure 7 sensors-18-04296-f007:**
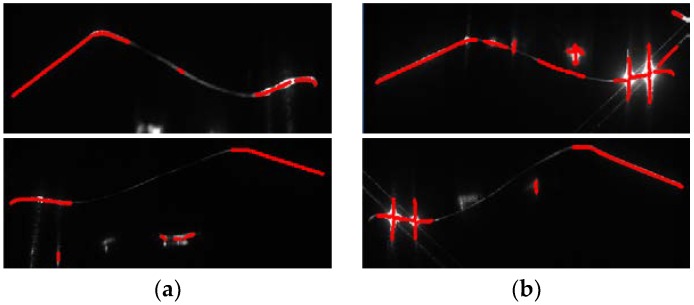
Center extraction results of stripe images. (**a**) Stripe images with low-brightness segments; (**b**) stripe images with overexposed segments.

**Figure 8 sensors-18-04296-f008:**
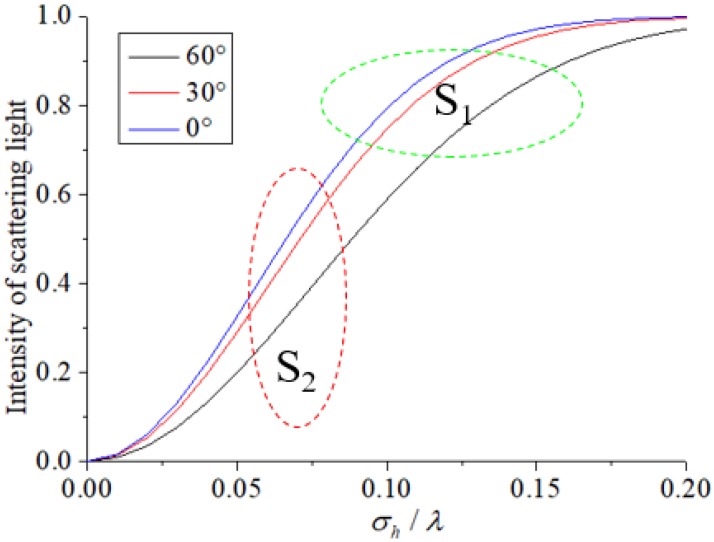
Scattering intensity as the ratio of the surface roughness and wavelength.

**Figure 9 sensors-18-04296-f009:**
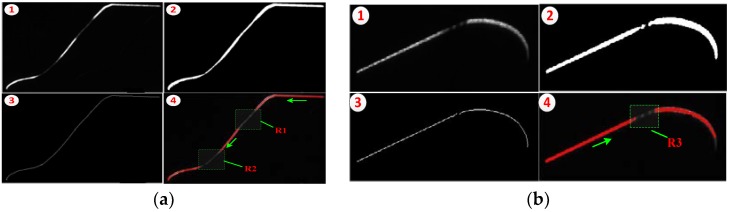
Location of stripe segments that are to be enhanced. (**a**) Images captured by the outer camera, (**b**) inner images. ① is the original image, ② shows the segmented and binarized image, ③ presents the stripe skeleton, and ④ indicates the regions established through stripe quality evaluation.

**Figure 10 sensors-18-04296-f010:**
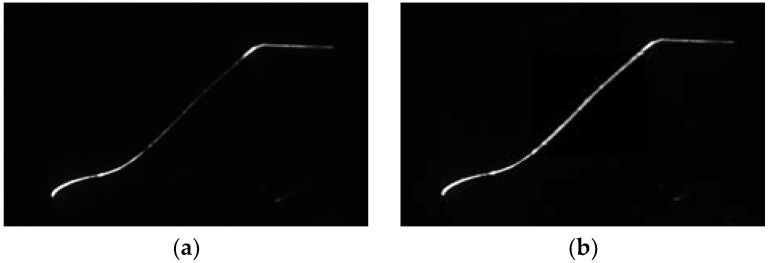
Stripe images before and after brightness enhancement. (**a**) Stripe image before enhancement; (**b**) stripe image after enhancement.

**Figure 11 sensors-18-04296-f011:**
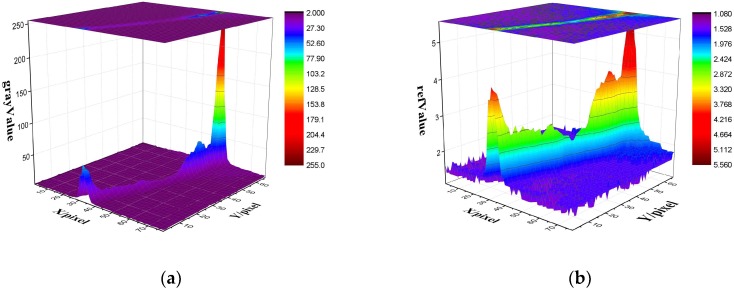
Stripe image gray and reflectivity distribution. (**a**) Gray distribution of the stripe image; (**b**) reflectivity distribution of the stripe image.

**Figure 12 sensors-18-04296-f012:**
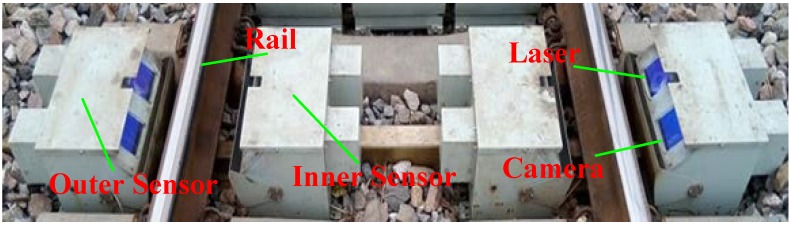
Wheel size measurement system in an outdoor complex environment.

**Figure 13 sensors-18-04296-f013:**
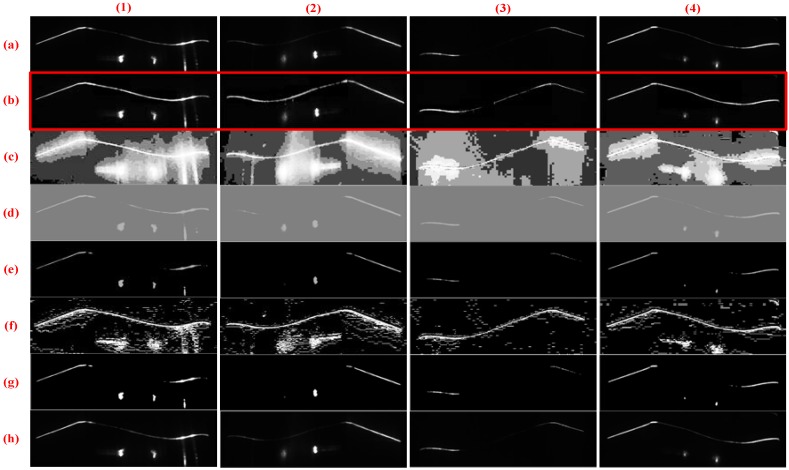
Enhanced results of outer stripe images based on different methods. Columns (**1**)–(**4**) present the tread segments with different missing percentages. Row (**a**) displays the raw images. Rows (**b**–**h**) exhibit the results processed by HE, CA, LT, CF, MF, and SO methods, respectively, and row (**b**) shows the results generated through the proposed method.

**Figure 14 sensors-18-04296-f014:**
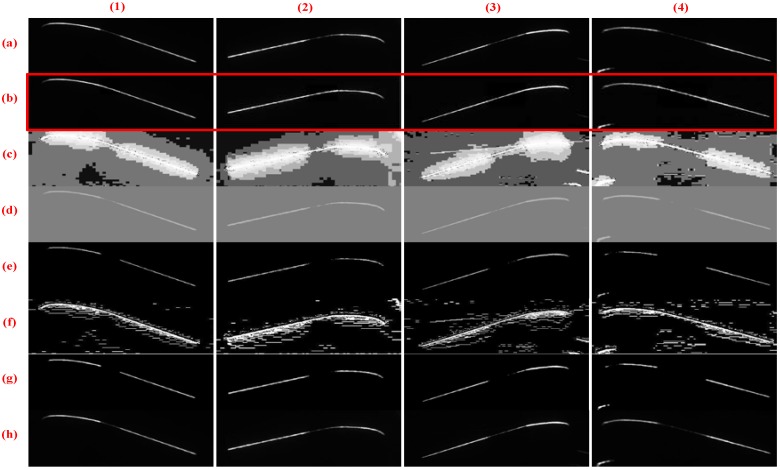
Enhanced results of inner stripe images based on different methods. Columns (**1**)–(**4**) present the tread segments with different missing percentages. Row (**a**) displays the raw images. Rows (**b**–**h**) exhibit the results processed by HE, CA, LT, CF, MF, and SO methods, respectively, and row (**b**) shows the results generated through the proposed method.

**Figure 15 sensors-18-04296-f015:**
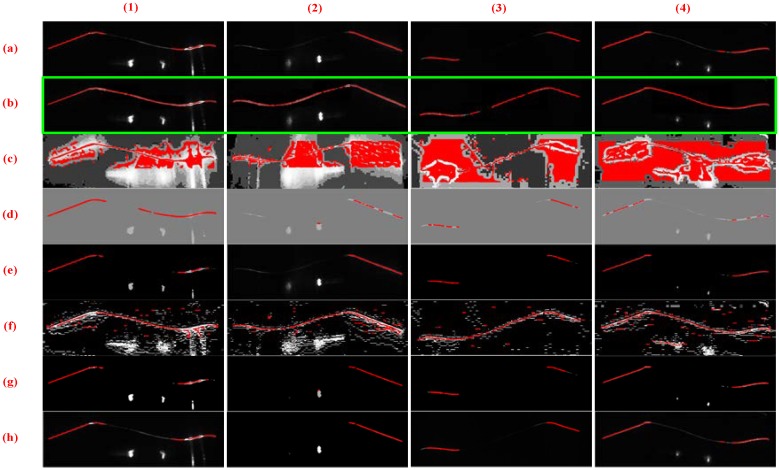
Center extraction results of outer stripe images based on different methods. Columns (**1**)–(**4**) present the tread segments with different missing percentages. Row (**a**) shows the raw images. Rows (**b**–**h**) display the center extraction results of images through HE, CA, LT, CF, MF, and SO methods, respectively, and row (**b**) exhibits the center extraction results of images processed through the proposed method.

**Figure 16 sensors-18-04296-f016:**
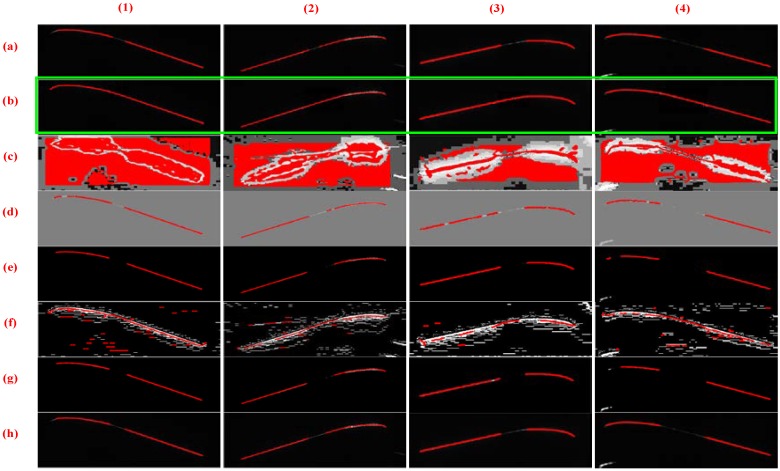
Center extraction results of inner stripe images based on different methods. Columns (**1**)–(**4**) present the tread segments with different missing percentages. Row (**a**) shows the raw images. Rows (**b**–**h**) display the center extraction results of images through HE, CA, LT, CF, MF, and SO methods, respectively, and row (**b**) exhibits the center extraction results of images processed through the proposed method.

**Figure 17 sensors-18-04296-f017:**
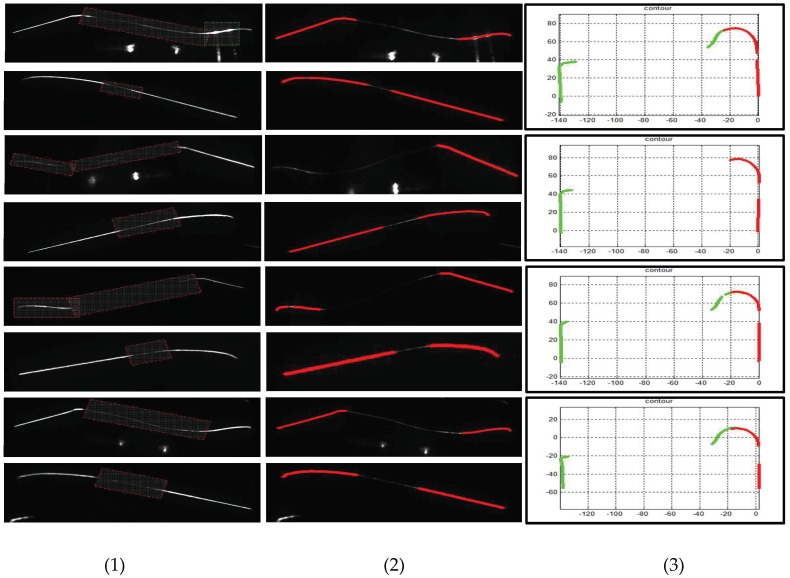
Wheel cross contour reconstruction without image processing. Column (**1**) presents the raw images. Column (**2**) shows the center extraction results of (1). Column (**3**) exhibits the wheel cross contour results of each wheel.

**Figure 18 sensors-18-04296-f018:**
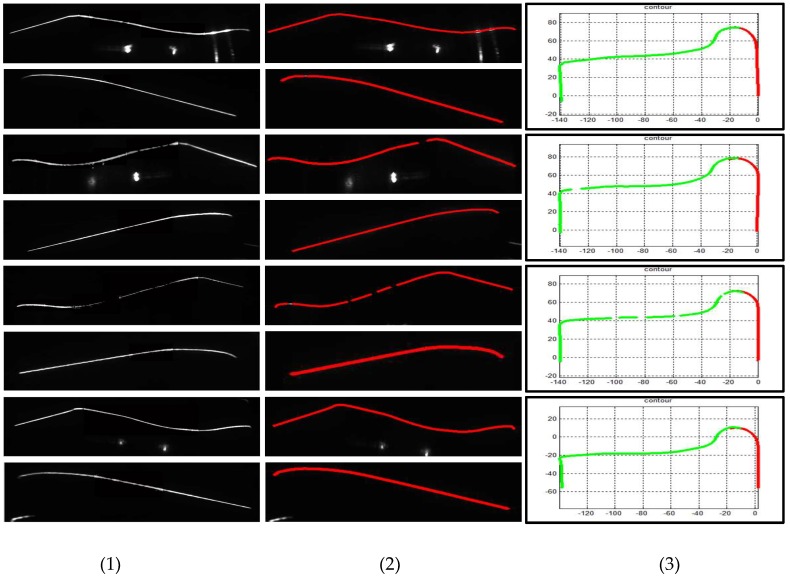
Wheel cross contour reconstruction with image processing. Column (**1**) presents the raw images. Column (**2**) shows the center extraction results of (1). Column (**3**) displays the wheel cross contour results of each wheel.

**Figure 19 sensors-18-04296-f019:**
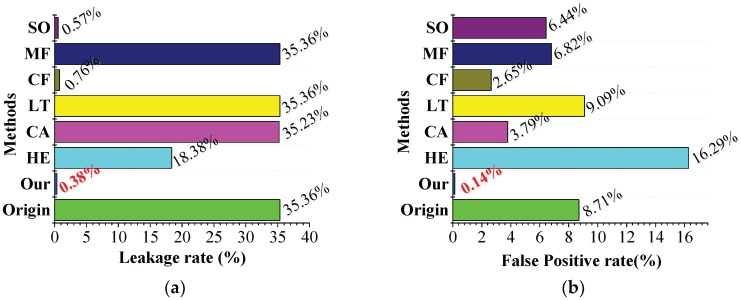
Leakage and false-positive rates of different methods in an outdoor complex environment. (**a**) Leakage rates. (**b**) False-positive rates.

**Table 1 sensors-18-04296-t001:** Calibration parameters of wheel sensors.

Wheel Sensors Parameters								
	f_x_	f_y_		u_0_	v_0_	k_1_		k_2_
Camera 1	1978.7510	1978.6804		680.5250	537.8380	−0.1383		0.1901
Camera 2	1971.5685	1972.6837		697.6733	536.6688	−0.1328		0.1476
	a		b		c		d	
PlaneFunc1	0.3190		0.8092		−0.4934		204.2166	
PlaneFunc2	0.1931		−0.8568		0.4782		−196.3384	
**T** _12_	**R** = $\left[\begin{array}{ccc} -0.4728 & -0.4973 & -0.7275\\ 0.6178 & 0.4016 & -0.6760\\ 0.6283 & -0.7690 & 0.1173 \end{array}\right]$, **t** = $\left[\begin{array}{c} 333.1798\\ 279.5493\\ 348.4826 \end{array}\right]$							

**Table 2 sensors-18-04296-t002:** Stripe quality evaluation under different processing methods.

Data	Methods	$\mu_{g}$	$\sigma_{g}$	$\mu_{\rho }$	$\sigma_{\rho }$	$D_{p}$	$\mu_{d}$	$\sigma_{d}$
DATA1	Origin	0.767	0.230	0.060	0.176	62.016%	0.142	0.214
	LT	0.671	0.130	0.053	0.154	47.651%	0.221	0.294
	MF	0.829	0.179	0.064	0.179	48.756%	0.233	0.271
	SO	0.749	0.220	0.058	0.170	60.221%	0.215	0.250
	Our	0.769	0.153	0.063	0.170	98.480%	0.024	0.009
DATA2	Origin	0.820	0.184	0.007	0.023	53.280%	0.116	0.172
	LT	0.673	0.090	0.007	0.027	51.181%	0.076	0.181
	MF	0.882	0.140	0.007	0.025	50.524%	0.091	0.198
	SO	0.794	0.171	0.007	0.024	52.624%	0.116	0.179
	Our	0.807	0.131	0.008	0.027	99.868%	0.008	0.0
DATA3	Origin	0.838	0.180	0.008	0.020	38.523%	-----	-----
	LT	0.689	0.082	0.008	0.029	34.496%	0.014	0.0
	MF	0.939	0.133	0.009	0.026	35.167%	0.010	0.0
	SO	0.818	0.187	0.008	0.020	37.583%	-----	-----
	Our	0.690	0.188	0.005	0.017	96.004%	0.016	0.012
DATA4	Origin	0.590	0.191	0.005	0.026	53.667%	0.673	0.0
	LT	0.582	0.228	0.007	0.041	34.704%	0.344	0.4743
	MF	0.676	0.228	0.004	0.027	35.241%	0.233	0.384
	SO	0.563	0.178	0.005	0.028	53.846%	0.677	0.0
	Our	0.564	0.264	0.006	0.028	90.595%	0.038	0.038

Note: “-----” indicates that the result is calculated unsuccessfully.

**Table 3 sensors-18-04296-t003:** Time statistics for stripe image processing (ms).

Methods	SD	SE	SS	SCE	TBA	Levels	AC
HE	----	28	----	520	548	18	30
CA	----	15	----	427	442	14	31
LT	----	29	----	71	100	4	25
CF	----	90	----	57	147	5	30
MF	----	14	----	24	38	2	19
SO	----	24	----	27	51	2	26
PM	222	137	32	30	421	13	33

Note: “----” indicates that the item is missing.

**Table 4 sensors-18-04296-t004:** Measurement accuracy of wheel size by using different methods (mm).

Processing Methods	Flange Thickness	Tread Wear	Rim Thickness
WP	------	------	------
PM	0.17	0.14	0.56
HE	5.82	5.40	8.28
CA	------	------	------
LT	------	------	------
CF	6.62	2.51	2.40
MF	------	------	------
SO	6.20	5.16	2.00

Note: “WP” stands for “without processing” and “------” indicates that the result is calculated unsuccessfully.

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
