# Peer review of "Reliable and Accurate Wheel Size Measurement under Highly Reflective Conditions"

_sensors, 2018, doi:10.3390/s18124296_

Round 1
Reviewer 1 Report
This paper propose mature solution for train wheel measurement. Main novelty of this paper is connected with image enhancement in highly reflective environment. Worth mentioning is that developed system has been tested in real cases.
I have some recommendations to improve the paper quality (especially important is point 5):
1) Fig. 1. The images presented are not High dynamic images. The are just images acquired in presented conditions. Please also show them in more detailed way. Some zoom and lines cross-sections to show readers details of intensity distribution. One or two examples are enough.
2) Fig. 5c could you show all cross-sections in the same scale in intensity axis? Then it will be easier to understand the 5a image. The quality of images should be improved also. They are too blurred and with low contrast.
3) Fig. 6 and 7: Could you go through the same examples of images with different processing? It will be easier to reader to follow. For example in Fig. 7 red curves are on the intensity values and it is very hard to understand it.
4) Fig. 11. The black background is not clear. Please provide better visualization of intensity distribution.
5) Section 3.3 is naïve from information theory point of view. Please remove it from paper and from abstract.
6) Table 3. Change “Meshods” to Methods.
Author Response
Please see responses as attached.

Reviewer 2 Report
Dear Authors,
Thank you for your contribution! The topic was very interesting and the paper was well written.
The method was clearly described and the result was promising. There has minor correction needed. Some figures are not clear: e.g. Figure 5, 8. Some figures didn't clarify the unit in the axis: e.g. Figure 11. Fonts are not consistent in figures and tables: Figure 2, 4, 7, 21, and Table 1, as ell as Equation: Equation (1).
Author Response
Please see responses as attached.

Round 2
Reviewer 1 Report
Dear Authors, I accept your comments and changes and recommend your work for publication.